# Tracing the Roots: Leveraging Temporal Dynamics in Diffusion Trajectories for Origin Attribution

**Andreas Floros**
Imperial College London
andreas.floros18@imperial.ac.uk

**Seyed-Mohsen Moosavi-Dezfooli**[*]
Apple
smoosavi@apple.com

**Pier Luigi Dragotti**
Imperial College London
p.dragotti@imperial.ac.uk

## Abstract

Diffusion models have transformed image synthesis through iterative denoising, by defining trajectories from noise to coherent data. While their capabilities are widely celebrated, a critical challenge remains unaddressed: ensuring responsible use by verifying whether an image originates from a model's training set, its novel generations or external sources. We introduce a framework that analyzes diffusion trajectories for this purpose. Specifically, we demonstrate that temporal dynamics across the entire trajectory allow for more robust classification and challenge the widely-adopted "Goldilocks zone" conjecture, which posits that membership inference is effective only within narrow denoising stages. More fundamentally, we expose critical flaws in current membership inference practices by showing that representative methods fail under distribution shifts or when model-generated data is present. For model attribution, we demonstrate a first white-box approach directly applicable to diffusion. Ultimately, we propose the unification of data provenance into a single, cohesive framework tailored to modern generative systems.

## 1 Introduction

Generative modeling has seen major advancements with the rise of diffusion models (Ho et al., 2020). These models have become the standard for image synthesis, achieving state-of-the-art performance via trajectories from noise to coherent data (Karras et al., 2024). However, their growing prevalence has raised significant concerns about privacy, security and accountability. For instance, sensitive or copyrighted data used during training can be memorized and inadvertently reproduced by the model during inference (Carlini et al., 2023; Somepalli et al., 2023; Gu et al., 2025). Furthermore, the lack of mechanisms to attribute content to specific models has made it easier to distribute harmful or malicious outputs without repercussions (Wang et al., 2023; Laszkiewicz et al., 2024; Liu et al., 2025). We are therefore broadly concerned with the following question regarding data provenance:

*Given pretrained generative models and data points, what relationships, if any, exist between them?*

Specifically, toward responsible generative modeling, our goal is to determine whether an image is (i) part of a model's training (*member*) set, (ii) a novel sample from the model (*belonging*) or (iii) sourced externally (*external*). In doing so, we propose to unify two traditionally separate tasks: Membership Inference Attacks (MIAs), which determine whether an image was part of a model's training set (Shokri et al., 2017; Yeom et al., 2018), and Model Attribution (MA), which determines whether an image was generated by a specific model (Wang et al., 2023; Laszkiewicz et al., 2024).

---

[*]Work done while at Imperial.

39th Conference on Neural Information Processing Systems (NeurIPS 2025).

We begin our exploration of data forensics by first focusing on the MIA setting. The literature on this task for diffusion models has largely converged to a likelihood thresholding approach, where the model's (negative) denoising loss is used as a proxy (Matsumoto et al., 2023; Duan et al., 2023). We identify two significant limitations with methods adopting this framework that, arguably, render them unworkable for practical membership inference. In particular, we show that such MIAs cannot distinguish between model-generated and training data, limiting their applicability in auditing synthetic data (e.g., for identifying memorization) or data extraction attacks. More importantly, we show that such threshold-based approaches may also underperform compared to naive baselines, that have no access to the underlying diffusion model and hence no real predictive power. We therefore argue that existing MIAs are, at best, strong *non-membership* inference attacks (Carlini et al., 2022).

The above-mentioned shortcomings of diffusion MIAs motivate us to depart from conventional approaches. Specifically, we challenge the widely-adopted and influential "Goldilocks zone" conjecture of Carlini et al. (2023), which posits that representations at the intermediate steps of denoising diffusion are most effective for membership inference. Intuitively, we expect that there is a wealth of information that is encoded within diffusion trajectories, and we hypothesize the existence of hidden patterns in the temporal dynamics of diffusion that may be used for more robust classification.

Equipped with our proposed trajectory representations, we then revisit the problem of data provenance. With our method, we demonstrate an application to data extraction, improved robustness to distribution shifts and a first white-box model attribution method that is directly applicable to diffusion models.

## 2 Background

We give a brief overview of diffusion and the various data provenance tasks we are concerned with. We refer the reader to Section 7 for a discussion on our work's position within the broader literature.

### 2.1 Diffusion-based generative models

**Continuous-time formulation**  Diffusion defines a mapping between the data distribution, $p$, and a tractable distribution. Let $\boldsymbol{x}(\cdot) : [0, T] \to \mathbb{R}^D$ such that $\boldsymbol{x}(0) \sim p$ and $\boldsymbol{x}(T)$ is normal. Specifically, for time $t = 0$ to $t = T$, we write the Itô Stochastic Differential Equation (SDE) (Song et al., 2021b):

$$\mathrm{d}\boldsymbol{x} = \boldsymbol{f}(\boldsymbol{x}, t)\mathrm{d}t + g(t)\mathrm{d}\boldsymbol{w}, \tag{1}$$

where $\boldsymbol{f}(\cdot, t) : \mathbb{R}^D \to \mathbb{R}^D$, $g(\cdot) : \mathbb{R} \to \mathbb{R}$ are appropriate drift, diffusion coefficients and $\boldsymbol{w}$ is the standard Wiener process. Given the above forward process, sampling is performed from $t = T$ to $t = 0$, by modeling trajectories with a reverse-time SDE (Anderson, 1982) as:

$$\mathrm{d}\boldsymbol{x} = [\boldsymbol{f}(\boldsymbol{x}, t) - g(t)^2 \nabla_{\boldsymbol{x}} \log p_t(\boldsymbol{x})]\mathrm{d}t + g(t)\mathrm{d}\bar{\boldsymbol{w}}, \tag{2}$$

where $\mathrm{d}t < 0$ and $\bar{\boldsymbol{w}}$ is the time-reversed standard Wiener process. Crucially, under this interpretation, modeling the reverse diffusion trajectories requires knowledge of the score function $\nabla_{\boldsymbol{x}} \log p_t(\boldsymbol{x})$.

**Denoising diffusion probabilistic models**  In practice, one discretizes the above equations and considers families of SDEs with a tractable forward process. A common parameterization is given by DDPMs (Ho et al., 2020), which choose $\boldsymbol{f}(\boldsymbol{x}, t) = -\frac{1}{2}\beta(t)\boldsymbol{x}$ and $g(t) = \sqrt{\beta(t)}$, where the function $\beta(t)$ is some noise schedule. By discretizing (1) with DDPM, the forward probabilities become:

$$p_t(\boldsymbol{x}_t | \boldsymbol{x}) = \mathcal{N}(\boldsymbol{x}_t; \sqrt{\bar{\alpha}_t}\boldsymbol{x}, (1 - \bar{\alpha}_t)\boldsymbol{I}), \tag{3}$$

where the subscripts denote discretization, $\alpha_t = 1 - \beta_t$ and $\bar{\alpha}_t = \prod_{s=0}^{t} \alpha_s$. As $p_t(\boldsymbol{x}_t | \boldsymbol{x})$ is normal, the score may be approximated by application of Tweedie's formula (Efron, 2011):

$$\nabla_{\boldsymbol{x}_t} \log p_t(\boldsymbol{x}_t) = \frac{\sqrt{\bar{\alpha}_t}\mathbb{E}(\boldsymbol{x}|\boldsymbol{x}_t, t) - \boldsymbol{x}_t}{1 - \bar{\alpha}_t}, \tag{4}$$

where the expectation describes the minimum mean squared error Gaussian denoiser. Therefore, by (3) and (4), Denoising Score Matching (DSM) may be performed via a noise-predicting neural network, $\boldsymbol{\epsilon_\theta}(\cdot, t) : \mathbb{R}^D \to \mathbb{R}^D$, that minimizes the quantity $\mathcal{L}_t = \|\boldsymbol{\epsilon_\theta}(\sqrt{\bar{\alpha}_t}\boldsymbol{x} + \sqrt{1 - \bar{\alpha}_t}\boldsymbol{\epsilon}, t) - \boldsymbol{\epsilon}\|_2^2$ for all $t$. The complete DDPM optimization objective is then expressed as follows:

$$\min_{\boldsymbol{\theta}} \mathbb{E}_{t \sim \mathcal{U}(\{0, \dots, T-1\}), \boldsymbol{x} \sim p, \boldsymbol{\epsilon} \sim \mathcal{N}(\boldsymbol{0}, \boldsymbol{I})} \lambda_t \mathcal{L}_t. \tag{5}$$

For appropriate $\lambda_t$, the above is equivalent to the Negative Evidence Lower Bound (NELBO) of the data (Ho et al., 2020). In this sense, DSM may also be reframed as likelihood maximization.

## 2.2 Data provenance

**Membership inference**  The aim of this task is to determine whether a given data point was used in the training of a machine learning model, i.e., whether it is a member. Traditionally, one assumes knowledge of the overall data distribution, $p$, but no knowledge of the specific training data. Moreover, other training details of the model are also assumed. Given this setup, Shokri et al. (2017) proposed to develop MIAs via surrogates, whose aim is to closely reproduce the target model without knowledge of its member set. As these surrogate models are trained by us, the attacker, it is then possible to optimize the MIA in a supervised way, hoping that it will also be transferable to the target model.

Since then, follow-up work by Yeom et al. (2018) proposed MIAs via loss thresholding, where the intuition is that training data will naturally achieve a smaller model loss compared to unseen data. In practice, this approach simplifies the design space of the attacks, making them largely parameter-free.

**Model attribution**  With the rise of increasingly capable generative models, the attribution of synthetic data has become a problem of interest to the research community (Wang et al., 2023; Laszkiewicz et al., 2024; Liu et al., 2025). Given such synthetic data, the aim here is to identify the *specific* generative model that is responsible for producing it. Arguably, assuming the generators adequately capture the underlying data such that there are no noticeable distribution shifts, this is a task that may require white-box access to the systems for more reliable attribution. Representative works operating under this assumption have converged to reconstruction or inversion-based methods, where the idea is to estimate model-specific, internal preimages of the samples that are more informative and distinguishable than the final outputs (Wang et al., 2023; Laszkiewicz et al., 2024).

# 3   Threat model

We frame data forensics as a game between model developers, whose interest is to minimize liability, and an adversary wishing to uncover data and its origins. The adversary's goal is to extract meaningful features, $\boldsymbol{f}$, that couple the diffusion model, $\boldsymbol{\epsilon_\theta}$, and the data, $\boldsymbol{x}$, in a way that reveals their underlying relationship. To this end, we will consider a simple pipeline consisting of a feature extraction stage from an off-the-shelf diffusion model and a learning stage. For a simple and standardized evaluation, we will adopt a linear probing model for classification to assess the quality of feature representations:

$$\boldsymbol{l}(\boldsymbol{x}; \boldsymbol{\epsilon_\theta}) = \boldsymbol{W} \cdot \boldsymbol{f}(\boldsymbol{x}; \boldsymbol{\epsilon_\theta}) + \boldsymbol{b}. \qquad (6)$$

Here, $\boldsymbol{l}$ represents logits for binary or ternary classification, depending on the task we consider, and $\boldsymbol{W}, \boldsymbol{b}$ are parameters to be optimized based on the adversary's capabilities. For clarity, we provide complete implementation details in Appendix A and state such capabilities explicitly below:

- *White-box access.* Given well-trained generative systems, we argue that black-box methods will struggle to distinguish the various classes of data under a fair evaluation with no distribution shifts, necessitating the use of internal model signals for reliable classification.
- *Limited data access.* Surrogate development, as proposed by Shokri et al. (2017), is often unrealistic and the alternative of method refinement at evaluation leads to inflated metrics. To properly develop forensics tools, we instead explicitly assume access to a small, potentially imbalanced or otherwise uncurated fraction of the member set (we will use $< 3.4\%$).

Note, the above-described threat model makes a non-standard assumption regarding data access and trades this for the surrogate framework of Shokri et al. (2017). Our position is that, while ours is a strong assumption, it is far milder than the alternative. We give two reasons to defend our choice.

On a practical level, we argue that surrogate development is computationally prohibitive for modern generative systems, especially since competitive MIAs operating under this framework may train several such surrogates to approximate the target's behavior (Carlini et al., 2023; Pang et al., 2025).

More fundamentally, under the surrogate framework, it is difficult to justify or bound our assumptions. In particular, to reproduce modern generative systems, a detailed description of the training setup may be required. However, important details such as pretraining, dataset composition, number of iterations and regularization are often omitted or otherwise difficult to infer even in open-source releases. We therefore argue that data provenance via surrogate development involves several hidden and intractable assumptions that make it harder to judge real data privacy and security risks.

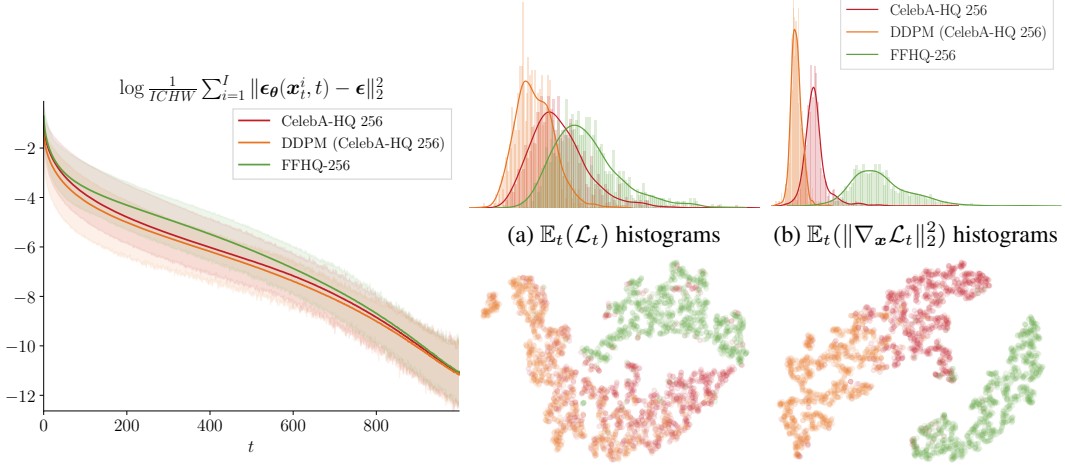

Figure 1: Member, model-generated and external data loss (averages with solid lines) as a function of $t$ for CelebA-HQ DDPM.

(a) $\mathbb{E}_t(\mathcal{L}_t)$ histograms

(b) $\mathbb{E}_t(\|\nabla_{\boldsymbol{x}}\mathcal{L}_t\|_2^2)$ histograms

(c) $\{\mathcal{L}_t\}_{t=0}^{T-1}$ t-SNE

(d) $\{\|\nabla_{\boldsymbol{x}}\mathcal{L}_t\|_2^2\}_{t=0}^{T-1}$ t-SNE

Figure 2: Visualization of CelebA-HQ DDPM features.

Table 1: TPRs / FPRs for MIA using (7). Top are FPRs on external and bottom are FPRs on model-generated data. The latter (TPR, FPR) are below the $y = x$ curve, i.e., worse than random.

| Dataset | $t = 50$ | $t = 100$ | $t = 150$ | $t = 200$ | $t = 250$ | $t = 300$ |
|---|---|---|---|---|---|---|
| CIFAR-10 | 56.3 / ⁴⁶·⁴/₆₀.₇ | 60.1 / 44.7/64.3 | 61.1 / 41.7/66.6 | 62.2 / 42.8/66.6 | 62.2 / 42.4/67.3 | 59.6 / 45.6/65.7 |
| CelebA-HQ 256 | 64.1 / 48.3/92.4 | 72.9 / 41.5/94.3 | 78.9 / 34.2/95.7 | 83.5 / 31.0/96.5 | 85.1 / 29.2/96.4 | 85.9 / 29.2/96.7 |

## 4 Do membership inference attacks work?

It is instructive to first review existing approaches to MIAs on diffusion models. Under the assumptions established in Section 3, these attacks are, arguably, useful only as filters for data extraction (Zhang et al., 2025). However, we will demonstrate that MIAs fail in the presence of model-generated data, limiting their potential in auditing model outputs or extracting training data. Even worse, we will highlight a further weakness and show how, under certain conditions, existing MIAs may underperform relative to naive baselines that provably have no real predictive power.

To this end, we consider the representative threshold-based approach of Yeom et al. (2018), which has inspired several recent works on diffusion MIAs (Carlini et al., 2023; Duan et al., 2023; Matsumoto et al., 2023; Kong et al., 2024; Tang et al., 2024; Zhai et al., 2024). For a given time-step, $t$, the idea is to identify as members the samples that attain a denoising loss, $\mathcal{L}_t$, below a certain threshold $\tau$:

$$\mathcal{M}(\boldsymbol{x}; \boldsymbol{\epsilon_\theta}) = \mathbb{1}[\mathcal{L}_t < \tau], \quad \mathcal{L}_t := \|\boldsymbol{\epsilon_\theta}(\sqrt{\bar{\alpha}_t}\boldsymbol{x} + \sqrt{1 - \bar{\alpha}_t}\boldsymbol{\epsilon}, t) - \boldsymbol{\epsilon}\|_2^2, \quad \boldsymbol{\epsilon} \sim \mathcal{N}(\boldsymbol{0}, \boldsymbol{I}). \quad (7)$$

Typically, $\tau$ is chosen to demonstrate the True Positive Rate at a set False Positive Rate (TPR @ FPR) or adjusted for other metrics, e.g., the Area Under the Curve (AUC) or Attack Success Rate (ASR). $t$ is reported to be a critical hyperparameter (Carlini et al., 2023) and commonly tuned via surrogates.

### 4.1 Representative attacks cannot audit synthetic data

For a DDPM trained on CelebA-HQ (Karras et al., 2018a) at $256 \times 256$ resolution, we plot $\mathcal{L}_t$ in Figures 1, 2a. It is clear that thresholding is not sufficient for reliable classification as synthetic data consistently fools such detectors by attaining the lowest loss. We confirm our findings with further experiments in Table 1 and also include a CIFAR-10 (Krizhevsky, 2009) DDPM in our analysis. Fundamentally, we attribute this phenomenon to the maximum likelihood interpretation of DSM, where the loss is equivalent to the NELBO of the DDPM (Ho et al., 2020). As hinted in Figure 2, this limitation motivates us to explore more separable features, beyond thresholding, later in Section 5.

Table 2: MIA (member-external) AUC, TPRs @ 1% FPR and ASRs. For the CIFAR-10 experiment, we use CIFAR-10.1 as the external set, where the naive baseline is close to random guessing and there are no distribution shifts. The CelebA-HQ experiment uses FFHQ as external data, where there are clear distribution shifts, as reflected by the baseline's performance. In the latter case, the representative threshold-based approaches of Matsumoto et al. (2023); Kong et al. (2024) underperform.

| Method | CIFAR-10 | | | CelebA-HQ 256 | | |
|---|---|---|---|---|---|---|
| | AUC | TPR @ 1% FPR | ASR | AUC | TPR @ 1% FPR | ASR |
| Naive (model-blind) | 52.2 | 0.0 | 52.0 | 94.4 | 60.1 | 86.6 |
| Matsumoto et al. (2023) | 63.2 | 3.3 | 59.7 | 85.2 | 26.4 | 76.2 |
| Kong et al. (2024) (PIA) | 66.9 | 5.1 | 62.4 | 62.5 | 0.1 | 58.1 |
| Pang et al. (2025) (GSA$_2$) | **82.7** | **12.5** | **73.5** | **100.0** | **99.6** | **92.5** |

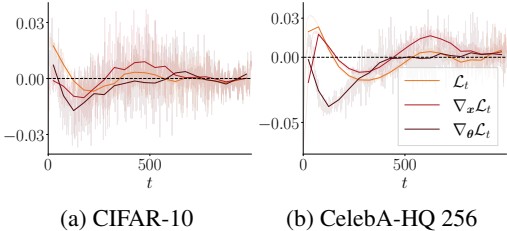

(a) CIFAR-10          (b) CelebA-HQ 256

Figure 3: Parameters from $\boldsymbol{W}$ in (6) corresponding to different features against $t$ for our MIAs.

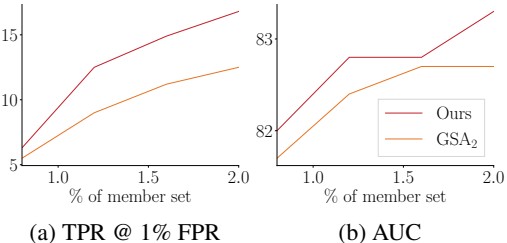

(a) TPR @ 1% FPR          (b) AUC

Figure 4: MIA performance as a function of the data budget (% of member set) on CIFAR-10.

Table 3: AUC for our MIAs (member-external) with $\{\mathcal{L}_t\}_{t=0}^{T-1}$ as features on CIFAR-10. We fix the number of queries to the diffusion model and consider different sampling strategies for $t$. We observe significant performance gains when the time-steps cover the entirety of the diffusion process.

| $t = 0, \ldots, 249$ | $t = 250, \ldots 499$ | $t = 500, \ldots, 749$ | $t = 750, \ldots, 999$ | $t = 0, 4, \ldots, 996$ |
|---|---|---|---|---|
| 68.5 | 68.1 | 54.7 | 50.7 | **71.2** |

## 4.2 Blind baselines may beat most membership inference attacks

We now focus on attacks under distribution shifts, where the member and external data may be distinguishable beyond the membership property.[1] For idealized external data, indistinguishable from the members barring membership, the literature often reports a random baseline, i.e., 1% TPR @ 1% FPR and 50% AUC, ASR. Here, we propose a potentially stronger, data-driven baseline that makes predictions without access to the diffusion model. In particular, we use a standard ResNet18 (He et al., 2016) classifier that is otherwise developed under identical conditions and with the same data budget as the other MIAs, i.e., 1000 members and 1000 external samples. Note, it is clear that such a naive method cannot possibly generalize, since the membership property is necessarily tied to the model. Therefore, it is expected that tailored MIAs, with access to the model, will outperform it.

To our surprise, however, the results of Table 2 reveal that the representative threshold-based attacks of Matsumoto et al. (2023); Kong et al. (2024) are brittle to distribution shifts. In particular, while we see reasonable performance on the CIFAR-10 DDPM when the external data is from CIFAR-10.1 (Recht et al., 2018; Torralba et al., 2008), the experiments on the CelebA-HQ DDPM with FFHQ (Karras et al., 2018b) external data show that the naive baseline significantly outperforms them.

Note, the superior attack of Pang et al. (2025), namely Gradient Subsampling and Aggregation (GSA), is not based on loss thresholding and instead computes a more involved feature vector as follows:

$$\boldsymbol{f}_i(\boldsymbol{x}; \epsilon_{\boldsymbol{\theta}}) = \mathbb{E}_t \|\nabla_{\boldsymbol{\theta}_i} \mathcal{L}_t\|_2^2, \quad \texttt{concat}(\ldots, \boldsymbol{\theta}_i, \ldots) = \boldsymbol{\theta}, \tag{8}$$

where $\boldsymbol{\theta}$ is partitioned into subsets $i$. As GSA maintains competitive performance, our findings suggest a failure specific to thresholding and further motivate us to investigate more robust representations.

---

[1]These are interclass shifts. There is also the possibility of intraclass shifts, e.g., the development budget consists of "cat" members, but at evaluation there exist "dog" members. See Table 4 for some analysis on this.

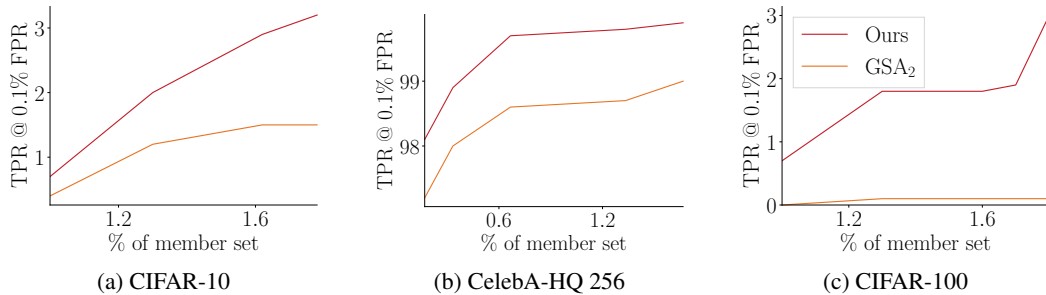

Figure 5: MIA TPRs @ 0.1% FPR as the data budget varies. See Appendix A for details.

Table 4: TPR @ 1% FPR for the MIA of Pang et al. (2025) (GSA$_2$) and our method ($\mathcal{L}_t, \nabla_{\boldsymbol{x}}\mathcal{L}_t, \nabla_{\boldsymbol{\theta}}\mathcal{L}_t$, $t = 0, \ldots, 999$). We report TPRs over ten random data splits that are not necessarily object-balanced, i.e., there may be intraclass distribution shifts. The mean and standard deviation are on the right.

| Method | CIFAR-10 | | | | | | | | | | $\mu \pm \sigma$ |
|---|---|---|---|---|---|---|---|---|---|---|---|
| GSA$_2$ | 9.7 | 9.6 | **11.7** | 8.0 | **9.8** | 8.2 | 7.1 | 8.8 | 11.1 | 6.6 | 9.1±1.64 |
| Ours | **18.3** | **10.9** | 11.3 | **14.5** | 9.4 | **11.9** | **11.3** | **13.0** | **12.4** | **7.4** | **12.0±2.93** |
| | CIFAR-100 | | | | | | | | | | |
| GSA$_2$ | 3.7 | 2.8 | 1.5 | 1.9 | 2.7 | 1.8 | 4.4 | 4.0 | 2.2 | 1.2 | 2.6±1.10 |
| Ours | **9.4** | **7.4** | **9.1** | **7.7** | **14.0** | **6.9** | **10.2** | **12.7** | **10.6** | **6.4** | **9.4±2.50** |

## 5 Rethinking the Goldilocks zone conjecture

*Conjecture* 1. If $t$ is too large, and so the noisy image is similar to Gaussian noise, then predicting the added noise is easy regardless if the input was in the training set; if $t$ is too small, and so the noisy image is similar to the original, then the task is too difficult. It is hypothesized that there exists a "Goldilocks zone" for membership inference (Carlini et al., 2023).

Our experiments in Section 4 justify an exploration of richer features for more robust classification, beyond brittle thresholding. Inspired by the GSA method of Pang et al. (2025), which extracts internal network signals via gradient information, and contrary to the influential "Goldilocks zone" hypothesis of Carlini et al. (2023), as stated in Conjecture 1, we propose to extract internal signals relating to the underlying diffusion processes. In particular, we will now explore vulnerabilities arising due to global temporal dynamics that would otherwise be discarded in previous approaches, hypothesizing that the paths traversed in high-dimensional ambient spaces are distinct for different classes of data.

Carlini et al. (2023) validated their conjecture on CIFAR-10 MIAs, where they found that $t \in [50, 300]$ is optimal. To motivate our approach, we revisit this setting and demonstrate alternatives that make use of temporal context. Specifically, given a fixed query budget to the diffusion model, we generalize the thresholding approach via concatenation of $\{\mathcal{L}_t\}_{t=0}^{T-1}$ into a feature vector meant to capture evolution.[2] The results using this proposed approach are in Table 3, with the setup of Section 4.2. Interestingly, while there exists a locally optimal region for $t$, in line with previous observations, we find that performance can be boosted significantly by uniformly distributing the time-steps over the entirety of the diffusion process. We therefore propose the counterhypothesis that there exist global patterns encoded in diffusion trajectories, containing valuable information for data provenance.

In particular, having freed ourselves from loss thresholding, we augment our MIAs with gradient features, $\{\|\nabla_{\boldsymbol{x}}\mathcal{L}_t\|_2^2, \|\nabla_{\boldsymbol{\theta}}\mathcal{L}_t\|_2^2\}_{t=0}^{T-1}$, with our complete feature extraction given in Algorithm 1 and visualizations in Figures 2, 3. Though we admittedly do not have a theoretical basis for including the gradients, we intuitively expect that they capture curvature information that is useful for trajectory modeling, given that $\mathcal{L}_t$ measures the score matching error. Moreover, as explored in Appendix B, we find they greatly improve our MIAs, making them competitive with GSA in Figures 4, 5 and Table 4.

---

[2]Note, our loss sequences are not native to DDPMs. The native forward process has independent Gaussian increments, with dependent overall noise at different $t$. Instead, our overall noise is completely independent of $t$.

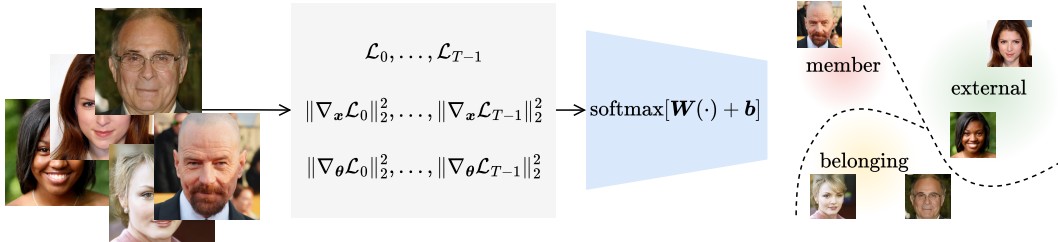

Figure 6: Our method. Given an image, we model its trajectory via features at different time-steps. We train classifiers on these representations to distinguish member, belonging and external data.

---

**Algorithm 1:** Trajectory feature extraction

**Input**: Data point $x$
**Parameters**: Trained DDPM, $\epsilon_{\boldsymbol{\theta}}$, with $\{\bar{\alpha}_t\}_{t=0}^{T-1}$
**Output**: Trajectory features $\boldsymbol{f}$
$\boldsymbol{f} \leftarrow \{\}$
**for** $t \in \{0, \ldots, T-1\}$ **do**
    $\epsilon \sim \mathcal{N}(\mathbf{0}, \boldsymbol{I})$
    $\mathcal{L}_t \leftarrow \|\epsilon_{\boldsymbol{\theta}}(\sqrt{\bar{\alpha}_t}x + \sqrt{1-\bar{\alpha}_t}\epsilon, t) - \epsilon\|_2^2$
    $\boldsymbol{f} \leftarrow \boldsymbol{f} \cup \{\mathcal{L}_t, \|\nabla_{\boldsymbol{x}}\mathcal{L}_t\|_2^2, \|\nabla_{\boldsymbol{\theta}}\mathcal{L}_t\|_2^2\}$
**return** $\boldsymbol{f}$

Table 5: Hyperparameters of our linear classifier, defined in (6). We refer the reader to Appendix A for further implementation details.

|        |                     |         |
|--------|---------------------|---------|
|        | batch_size          | 50      |
|        | epochs              | 100     |
| AdamW  | learning_rate       | 1e-3    |
|        | weight_decay        | 10      |
| StepLR | step_size           | 5 epochs |
|        | learning_rate_decay | 0.8     |

## 6  Toward origin attribution

Having investigated temporal dynamics in MIAs, we now generalize to Origin Attribution (OA), aiming to unify data provenance under a cohesive framework. Arguably, in this context, there are two fundamental relationships between models and data, i.e., *membership* and *belongingness*. To this end, we elevate binary MIAs into ternary classification, with our overall implementation summarized in Figure 6 and Table 5. In what follows, we conduct experiments toward this more ambitious goal.[3]

### 6.1  Stronger membership inference attacks

Here, we show how our ternary OA may address the limitation discussed in Section 4.1 by including model-generated data during classifier training, maintaining compliance with our stated assumptions in Section 3. With this enhancement, we develop a method suited for OA and benchmark its performance in Figure 8 and Table 6. Importantly, we also demonstrate an application to the more ambitious task of data extraction. Specifically, by inspecting our system's misclassifications, we collect a subset of model generations that are classified as members and investigate their similarity to the training data of the model. We report our findings with this approach in Figure 7, where we filter 30k model-generated samples down to 1.7k that resemble members, as quantified via the SSCD score (Pizzi et al., 2022). With reference to Figure 7a, it is clear that similarity is left-skewed for such misclassified data, validating our method.

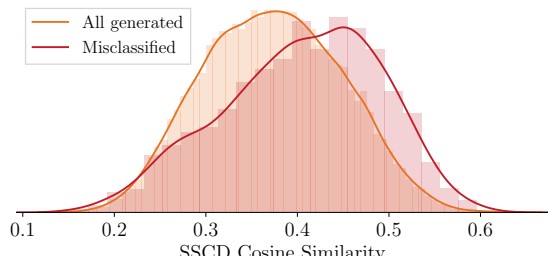

(a) Similarity histograms of generated images to members

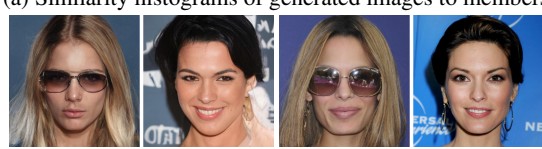

(b) Misclassified, generated    (c) Similar members

Figure 7: We investigate generated data that is misclassified by our system and identify similar samples to members (all unseen by the classifier in training).

---

[3] Following Wang et al. (2023), we refer to novel model-generated data as *belonging*, noting potential overlap of generated data and members when models memorize. Also note that we have overloaded their OA term.

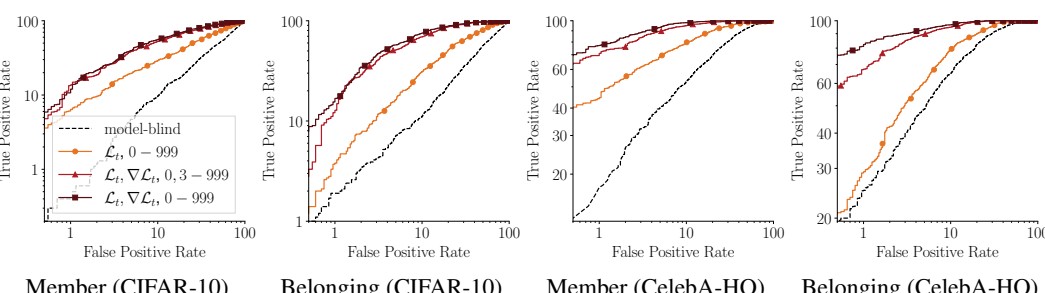

Figure 8: OA ROCs, taking member or belonging classes as positives.

Table 6: Average AUC, TPRs @ 1% FPR and ASRs for OA methods (member-belonging-external).

| Method | | | | CIFAR-10 | | | CelebA-HQ 256 | | |
|---|---|---|---|---|---|---|---|---|---|
| | | | | AUC | TPR @ 1% FPR | ASR | AUC | TPR @ 1% FPR | ASR |
| Naive (model-blind) | | | | 51.5 | 1.2 | 34.7 | 90.0 | 32.1 | 74.4 |
| **Our method** | | | | | | | | | |
| time-steps | $\mathcal{L}_t$ | $\nabla_x \mathcal{L}_t$ | $\nabla_\theta \mathcal{L}_t$ | | | | | | |
| $0, 1, \ldots, 999$ | ✓ | | | 71.6 | 6.9 | 54.9 | 95.2 | 52.7 | 84.4 |
| $0, 3, \ldots, 999$ | ✓ | ✓ | ✓ | 85.0 | 14.3 | 69.4 | 98.8 | 79.5 | 92.7 |
| $0, 1, \ldots, 999$ | ✓ | ✓ | ✓ | **86.5** | **15.9** | **71.1** | **99.3** | **86.5** | **94.1** |

## 6.2  Model attribution on diffusion models

Our analysis so far has centered around MIAs. We now specifically focus on attribution of synthetic data: Beyond membership, do trajectories also capture fingerprints useful for robust MA? To investigate, we train classifiers to distinguish DDPM, synthetic data and non-belonging, real data, with the same budget of 1000 external samples as in Section 4.2. For assessing robustness, we evaluate on non-belonging data produced by other generative models, i.e., DDIM (Song et al., 2021a), WaveDiff (Phung et al., 2023) and DDGAN (Xiao et al., 2022). In this sense, this is a fine-grained task aiming to extract features that are capable of attribution to a *specific* model. Crucially, other generators are unseen during development, reflecting an open-world setting (Laszkiewicz et al., 2024).

The results using the above-described setup are in Table 7. While our approach works overall, note it is not without limitations. In particular, we observe failure cases on DDGAN and DDIM samples, where a model-blind baseline may achieve higher conditional accuracy. However, referencing class-balanced accuracy in Table 7, the predictions of the model-blind classifiers are, overall, close to random guesses. We therefore caution that conditional accuracy is not sufficient to judge performance.

Actually, our analysis here leads us to question the viability of black-box methods for the task. Arguably, sufficiently capable generative models should produce samples perceptually indistinguishable from real data, motivating us to explore model-internal signals. Despite this, we note that there are recent works claiming promising results even in the black-box setting, by exploiting representations from foundational models (Liu et al., 2025). However, we deem such approaches to be methodologically impure and outside our scope as their performance may be attributed to the underlying pretraining. In particular, since foundation model development details are often opaque, it becomes difficult to assess effectiveness when a method's assumptions are unclear or otherwise unbounded.

It is also important to clarify that, while we are certainly not the first to explore the MA task in the white-box setting (Wang et al., 2023; Laszkiewicz et al., 2024), to our knowledge, we are the first to demonstrate white-box MA directly on diffusion. We stress that the above-mentioned works rely on reconstruction / inversion techniques that have only been shown to work *indirectly* on latent or distilled models (Rombach et al., 2022; Song et al., 2023). Specifically for the former, attribution is performed on autoencoders, saying nothing about the underlying diffusion model. Moreover, approaches based on reconstruction error may be fundamentally doomed in DDPMs, since perfect reconstruction is theoretically always possible, regardless of the sample. For example, see the inversion in Song et al. (2021a) and also of Huberman-Spiegelglas et al. (2024), which is applicable to stochastic processes.

Table 7: Performance of DDPM MA. We show per-model and average (class-balanced) accuracy. During development, classifiers only see DDPM-generated samples and non-belonging, real samples.

| Method | | | | CIFAR-10 | | | | | CelebA-HQ 256 | | | | |
|---|---|---|---|---|---|---|---|---|---|---|---|---|---|
| | | | | DDPM | DDIM | DDGAN | WDiff | Avg | DDPM | DDIM | DDGAN | WDiff | Avg |
| Naive (model-blind) | | | | 60.8 | 39.1 | **41.9** | 44.0 | 51.2 | 88.3 | **20.2** | 22.6 | 27.0 | 55.8 |
| **Our method** | | | | | | | | | | | | | |
| time-steps | $\mathcal{L}_t$ | $\nabla\mathcal{L}_t$ | | | | | | | | | | | |
| $0, 1, \ldots, 999$ | ✓ | | | 75.5 | 57.6 | 36.2 | 64.4 | 64.1 | 98.9 | 6.3 | 59.6 | 57.8 | 70.0 |
| $0, 3, \ldots, 999$ | ✓ | ✓ | | 86.0 | 72.4 | 18.1 | 76.1 | 70.8 | **100.0** | 7.9 | 74.2 | **76.9** | **76.5** |
| $0, 1, \ldots, 999$ | ✓ | ✓ | | **87.2** | **86.8** | 13.1 | **91.7** | **75.5** | **100.0** | 3.5 | **86.8** | 68.9 | **76.5** |

## 7 Discussion

We conclude with a discussion of our study's limitations and with recommendations for future research based on our findings. We have taken a first step in addressing general origin attribution. However, we believe that the task is far from solved, and we hope our work inspires the community to make further advancements toward more practical and robust data forensics tools.

### 7.1 Limitations

Our attacks' times are comparable to diffusion inference, making our approach significantly slower compared to existing membership inference attacks. However, as we are not concerned with real-time application, this is not unworkable. More importantly, in our model attribution experiments of Section 6.2, we observed failure cases that require further investigation. In general, our developed data provenance tools and evaluation should be made more robust, especially since our benchmarking was with controlled external sets. Other important aspects that warrant futher exploration include robustness to intraclass shifts, common and adversarial data corruptions as well as our approach's generalization ability and extensions to large, foundational generative systems.

### 7.2 Recommendations

**Revisit the threat models**   As we have argued in Section 3, the traditional surrogate approach to membership inference (Shokri et al., 2017) is not viable for modern generative systems. Beyond the issues relating to computational complexity, the core concern is the implicit and opaque assumptions that this framework entails, which are not usually met in practice. Granted, our proposed threat model also makes strong assumptions regarding data availability. However, we argue that ours are more practical, better defined and quantified. In the same spirit, for the sake of methodological purity, we also recommend against the use of foundational or otherwise pretrained models in the context of resource constrained and sensitive applications such as model attribution.

**Embrace distribution shifts**   It is important to contextualize metrics with suitable baselines. As also shown in Section 4.2, distribution shifts may lift such baselines from random guessing to seemingly high-performing solutions (Das et al., 2025). While there is recent work that explores data sanitization to mitigate these shifts (Dubiński et al., 2024), this approach risks discarding a still valuable, potentially informative part of the data and may end up crippling applicability. We therefore argue for an alternative, simpler protocol, where distribution shifts are embraced and performance must be judged relative to appropriate, naive baselines that are developed under a fair setting.

**Focus on data extraction**   Ultimately, as also discussed in Section 4, our position is that MIAs in the literature *cannot* prove membership, since they do not bound the FPRs. Specifically, under our stated assumptions in Section 3, Zhang et al. (2025) argue that membership inference is only useful as a subcomponent of a training data extraction attack (Carlini et al., 2023). Indeed, while we have also performed an evaluation via the standard AUC, TPR @ FPR and ASR metrics, we believe that it is our experiment in Figure 7 that is most insightful about real privacy and security risks. Going forward, we therefore advocate for the membership inference community to forgo the conventional benchmarks in favor of a more challenging and ambitious training data extraction task.

## Acknowledgments and Disclosure of Funding

Andreas acknowledges support from a NeurIPS Scholar Award. Seyed contributed to this work while at Imperial and remained involved after joining Apple. The authors declare no competing interests.

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

Table 8: Ablation study on the choice of features for our MIAs. We use $t = 0, 1, \ldots, 999$.

| | | | | CIFAR-10 | | | CelebA-HQ 256 | | |
|---|---|---|---|---|---|---|---|---|---|
| PIA | $\mathcal{L}_t$ | $\nabla_{\boldsymbol{x}}\mathcal{L}_t$ | $\nabla_{\boldsymbol{\theta}}\mathcal{L}_t$ | AUC | TPR @ 1% FPR | ASR | AUC | TPR @ 1% FPR | ASR |
| | ✓ | | | 73.3 | 6.5 | 68.1 | 99.1 | 95.4 | 96.4 |
| | | ✓ | | 74.9 | 6.1 | 68.5 | 99.3 | 97.5 | 97.3 |
| | | | ✓ | 77.0 | 3.1 | 71.6 | **100.0** | 99.5 | 93.7 |
| | ✓ | ✓ | | 80.0 | 12.2 | 72.4 | 99.7 | 99.2 | 98.9 |
| | ✓ | | ✓ | 80.5 | 10.5 | 72.8 | **100.0** | 99.9 | 98.4 |
| | | ✓ | ✓ | 81.8 | 11.1 | 74.1 | **100.0** | **100.0** | 98.9 |
| | ✓ | ✓ | ✓ | **83.3** | **16.8** | **74.8** | **100.0** | **100.0** | **99.5** |
| ✓ | ✓ | | | 69.5 | 5.2 | 64.2 | 99.2 | 89.4 | 95.5 |
| ✓ | | ✓ | | 75.6 | 1.0 | 68.7 | 88.1 | 51.3 | 80.7 |
| ✓ | | | ✓ | 71.7 | 3.3 | 64.5 | 97.1 | 40.2 | 92.8 |
| ✓ | ✓ | ✓ | | 75.3 | 5.2 | 68.6 | 99.3 | 89.6 | 95.7 |
| ✓ | ✓ | | ✓ | 72.9 | 9.8 | 66.8 | 99.2 | 89.0 | 96.3 |
| ✓ | | ✓ | ✓ | 75.6 | 1.6 | 68.6 | 97.5 | 46.0 | 93.3 |
| ✓ | ✓ | ✓ | ✓ | 76.3 | 6.1 | 69.1 | 99.3 | 90.0 | 96.4 |

## A   Experimental setup

Sampling and feature extraction from the generative models was conducted on a Linux cluster with Quadro RTX 6000 GPUs. We then developed our framework on a Windows laptop with a GTX 1650 GPU. All demonstrations are on off-the-shelf DDPMs, according to Ho et al. (2020). By default, we budget 1000 (object-balanced) samples per class for classifier development and evaluate on separate, similarly chosen datasets. CIFAR experiments use external sets (CIFAR-10.1/100 val) with minimal interclass shifts, whereas CelebA-HQ experiments use FFHQ as the external set with notable shifts. After feature normalization to zero mean and unit variance based on training data statistics, we fix the hyperparameters in Table 5 to optimize (6) via a cross-entropy loss. Hyperparameters are optimal (TPR @ 1% FPR on CIFAR-10) for Pang et al. (2025) and we did not perform a search beyond that.

## B   The choice of features

We experiment with different features for modeling diffusion trajectories. From Table 8, feature combination yields the best results. When integrating PIA (Kong et al., 2024), we replace $\epsilon \sim \mathcal{N}(\boldsymbol{0}, \boldsymbol{I})$ in $\mathcal{L}_t$ with $\epsilon_{\boldsymbol{\theta}}(\boldsymbol{x}, 0)$, making the features deterministic. However, we see that these features are not as performant as their stochastic counterparts. As a further exploration, we investigate whether trajectories can reveal the object-class of samples on CIFAR-10 in Table 9, i.e., standard classification. Interestingly, this approach is much better than random guessing ($\sim 10\%$).

Table 9: CIFAR-10 classification with trajectory features.

| time-steps | $\mathcal{L}_t$ | $\nabla_{\boldsymbol{x}}\mathcal{L}_t$ | $\nabla_{\boldsymbol{\theta}}\mathcal{L}_t$ | Accuracy |
|---|---|---|---|---|
| $0, 1, \ldots, 999$ | ✓ | | | 26.5 |
| $0, 3, \ldots, 999$ | ✓ | ✓ | ✓ | 31.4 |
| $0, 1, \ldots, 999$ | ✓ | ✓ | ✓ | **31.6** |

## C   Broader impact

The methods presented in this paper are developed to ensure accountability and transparency in scenarios involving open-weight models with undisclosed training data. By enabling data owners to determine whether their data was used in training a model, these techniques support ethical development, protect intellectual property rights and enhance privacy by identifying memorization. Furthermore, they allow for the attribution of harmful or malicious content to specific models, contributing to greater safety and fostering trust in these technologies. The methods also present potential risks, such as enabling the deanonymization of data, exploiting privacy vulnerabilities or unfairly penalizing applications due to inaccurate attributions.

