# OpenReview forum: "Tracing the Roots: Leveraging Temporal Dynamics in Diffusion Trajectories for Origin Attribution"
_NeurIPS.cc/2025/Conference — NeurIPS 2025 poster_

### Official Review · Reviewer_c42K · 2025-07-03

**Clarity:** 2
**Significance:** 2
**Originality:** 3
**Rating:** 4
**Confidence:** 3

**Summary:**

This paper endeavours to ensure responsible use of diffusion models and propose to unify membership inference and model attribution into a single framework - origin attribution (OA). It does this by identifying the limitations in the existing membership inference methods that existing methods fail under distribution shifts or when model-generated data is present, then propose to use characteristics in diffusion trajectory in DDPM to classify data origins. Experiments are conducted on DDPMs.

**Questions:**

1. Can you provide additional clarification regarding the practical significance of the assumptions this paper make (i.e., the use cases) as mentioned in the weaknesses section?
2. Would the proposed method works in more practical and widely used diffusion frameworks?
3. Can you provide more insights and analyse of how characteristics of diffusion trajectories are leveraged?
4. Can you show more experimental results to compare with current state-of-the-arts across all three tasks (MIA, MA, memorization) that this paper claims to tackle?

**Ethical Concerns:**

["NO or VERY MINOR ethics concerns only"]

**Final Justification:**

The authors have provided a detailed rebuttal containing additional clarifications and results. These have well-addressed my concerns, so I have raised my rating accordingly.

**Limitations:**

yes

**Quality:**

2

**Strengths And Weaknesses:**

Strength:

1. This paper tackles important issues that contribute to responsible AI. Specifically, it spans over the field of membership inference, model attribution, and memorization in diffusion models.
2. In addition, this paper works towards unifying the aforementioned field into a single, cohesive framework.
3. It is motivated by the gap in MIA in the literature that existing methods often fail for model-generated synthetic data, and the gap in MA that there is a lack of a white-box MA approach.

Weaknesses:

1. It would be better if more explanations regarding how often the assumed application scenario for the proposed general problem of origin attribution (OA) can be observed. As the authors acknowledged, the assumption is that the proposed methods only work when (1) model owners have not disclosed the training setup and data details, and (2) a small fraction of the training samples is available. I am concerned about the practical significance of the latter assumption, as it is mostly the case that users do not have a clue about the training samples used, let alone having a small fraction of them.
2. This paper delves into the privacy issue in the framework of DDPM. However, the most widely used diffusion models are often latent diffusion models, DiT models, flow-based models, etc. It would be interesting to see how the method performs in these, as their generations are more widespread, thus privacy protection is more practically significant.
3. Despite the motivation of unifying MIA and MA being well-supported, I find the method section less clearly explained, especially the section regarding the core insight of how characteristics of diffusion trajectories are leveraged. More explanations and analyses need to be conducted to make it more convincing.
4. For the MIA results, the paper only compares its performance with GSA to demonstrate effectiveness. For the other areas that this work also claims to tackle (MA and memorization), it is also suggested to provide more comprehensive comparisons to existing works. The experiments are quite insufficient in their current state.

---

> ### Author Rebuttal · Authors · 2025-07-28
>
> Thank you for your review. We are pleased that you appreciated our arguments for the unification of data forensics into a single framework. We address your concerns below:
>
> 1 Assumptions
> -
>
> (We repeat part of our answer to 13uc, who asked a similar question) **Our methodology is more scalable and practical than existing works (e.g. [1] [2], [3]) since we make far milder assumptions in our setup**. For example, as we mention in the paper, **in MIAs it is common to train shadow models to mimic the target DDPM, i.e., assume knowledge of training details of the diffusion models and knowledge of the member distribution to replicate the training.** In contrast, **we avoid this** and make our assumptions explicit in the form of a limited data budget, which is much more realistic and plausible. Of course, one would rather perform robust MIAs without any assumptions but, to our knowledge, no such method is known (and perhaps this might be theoretically impossible).
>
> **When do the assumptions hold & use cases?** As explained above, knowledge of part of the member set is a weaker and more plausible assumption than being able to replicate diffusion model training. To give an concrete example, **our assumptions hold in the case of the Stable Diffusion where it is known that LAION was used as the member set. In contrast, shadow model training is out of the question at such a scale.**
>
> Note, **even if the training set is (partly) known, simply searching through it becomes very challenging at large scales so our framework could be useful even in this context.**
>
> Furthermore, while we have mostly motivated data forensics from an outsider's perspective, **model developers may also want to develop such forensic tools to ensure fair use of their systems and to comply with regulations** (assumptions trivially hold in this case).
>
> 2 Demonstration in more diverse / larger settings
> -
>
> We are currently running more experiments (DiT, ImageNet). Due to resource and time constraints, it is unlikely that we will be able to complete these during rebuttal. However, we include benchmarking of our method and our strongest competitor, GSA [1], on CIFAR-100. Specifically, we train the diffusion model (with the flow-matching framework) on 50k samples of 100 different objects and set the data budget to 2% (1k). The external data is chosen from the 10k samples of the validation set of CIFAR-100, so we don't expect to see any distribution shifts. Again, **to our knowledge, our work demonstrates the largest scale MIAs, thoroughly evaluted under a controlled experimental setting with explicit and limited assumptions**:
>
> | CIFAR-100 MIA @ 2% budget | AUC        | TPR @ 1% FPR | ASR     |
> |---------------------------|------------|--------------|---------|
> | Matsumoto et al. [2]      | 58.9       | 1.4          | 56.7    |
> | Pang et al. (GSA) [1]      | 75.7       | 2.7          | 69.8    |
> | **Our method**            | **79.5**   | **10.8**     | **71.6**|
>
>
> We have also benchmarked performance in the case of object imbalance, simulating a realistic scenario where one is dealt a member set for training the classifier that is not representative of the member distribution. **In all of these experiments, trajectory features outperform the strongest competitor by a large margin** (see also Table 10 and Appendix D for further details and motivation regarding this setting):
>
> | CIFAR-100, TPRs @ 1%FPR | Our method     | GSA        |
> |-------------------------|----------------|------------|
> | Split 1                 | **7.2**        | 3.4        |
> | Split 2                 | **6.1**        | 2.4        |
> | Split 3                 | **7.9**        | 2.5        |
> | Split 4                 | **6.2**        | 3.6        |
> | Split 5                 | **11.4**       | 6.2        |
> | Split 6                 | **5.5**        | 2.1        |
> | Split 7                 | **6.4**        | 3.2        |
> | Split 8                 | **7.1**        | 4.6        |
> | mean ± std              | **7.2 ± 1.85** | 3.5 ± 1.35 |
>
> We stress that, even in these experiments, which are relatively small scale compared to something like LAION, it is clear that the data forensics tasks are far from solved and still remain highly non-trivial. **Specifically, as highlighted by recent works, [5], [6], the community is still far from reliably tackling data forensics in large scale models. Given that even at these relatively small scales robust data forensics remains a largely unsolved problem, our work demonstrates significant progress in the field that should not be discounted**.
>
> **Comments on standard diffusion vs more advanced constructions** As we also mention in the paper, **our focus on standard diffusion was intentional: it is the most widely studied and reproducible baseline**. Extending to latent/conditional models introduces confounding variables (e.g., VAEs, text encoders), which should be carefully treated to ensure a clean ablation and proper benchmarking (see [5], [6] and our discussion on robustness with belonging data and with distribution shifts, prior benchmarking is brittle as it fails to bound false positives). Besides, as standard diffusion remains at the core of these more advanced constructions, we believe our analysis and insights remain highly relevant.
>
> 3 How are characteristics of diffusion trajectories leveraged?
> -
>
> We will revise the method section and elaborate further. We include some comments here whcih we hope answer your question.
>
> Our main argument is that **there is rich information encoded in the diffusion trajectories that gets discarded by simple thresholding**. Prior work [4] focused on selecting a single time-step and they intuitively found that there is a Goldilocks zone for it where the best performance is achieved. Since then, the Goldilocks zone hypothesis and simple thresholding have become standard in diffusion data forensics. Our work disproves the hypothesis directly, as shown in Table 4 of the paper. Ultimately, the intuition is that **we exploit privacy and security vulnerabilities in the temporal dynamics of diffusion, not just the information at a single step. To get a sense of how the signals at different time-steps are leveraged by our classifiers, please see Figure 4 of the paper**. Notably, the weights of our classifiers are not localized in t, and we observe significant values spanning the majority of the trajectories.
>
> If you have any specific suggestions as to how we can make the intuition even more clear with additional experiments and analysis please do let us know and we will update the paper.
>
>
> 4 Comparison with SOTA
> -
>
> **We have included more experiments and comparisons in point 2 of our rebuttal above.**
>
> **Comparison with SOTA MIAs** Please note that **we have compared our MIAs with several existing works [1], [2], [3] as well as a black-box baseline (to quantify distribution shifts) in Table 5. Most methods adopt a threshold-based approach that we have repeatedly shown is not robust**, e.g. [2], [3], so we expect any other works that use the same ideas to perform similarly. **Notably, we consistenly outperform the top competitor [1] (in the paper and in additional experiments given above)**. If there is a specific work that you would like us to discuss we are happy to do so.
>
> **Comparison with MA methods: As you also acknowledge in the strengths section of your review, "there is a lack of a white-box MA approaches" for diffusion in the literature.** We have discussed this point in our rebuttal to HK2L's review and in the background section of our paper (starting at line 107, also see discussion starting at line 263). **To our knowledge, there is no existing white-box MA for diffusion.  As for black-box methods, you can see in Table 6 that we outperform them under a fair setting (top row).** Of course, if you are aware of existing white-box MA methods that work directly on diffusion models, please let us known and we would be happy to comment on them.
>
> **Comments on memorization search** We are not sure what you mean by comparing to SOTA in this category. To clarify, identifying memorized samples by filtering generated data is, to our knowledge, not a standard benchmark in membership inference literature. Actually, **as we experimentally prove in our paper (e.g. Figures 2, 3, Table 3), existing MIAs cannot be used for this purpose as they cannot distinguish between belonging and member data. The fact that our work enables this kind of application while existing MIAs completely fail precisely demonstrates the utility and practical use of our unified approach to data forensics**.
>
> ---
>
> We hope our responses have addressed your concerns and we believe that our additional experiments are convicing of the superiority and robustness of our approach compared to existing methods. We remain open to further discussion if there are any further clarifications needed. Please consider raising your score if you find the above answers sufficient.
>
> References
> -
>
> [1] Pang Y., et al., 2024, White-box Membership Inference Attacks against Diffusion Models
>
> [2] Matsumoto et al., 2023, Membership Inference Attacks against Diffusion Models
>
> [3] Kong F., et al., 2024, An Efficient Membership Inference Attack for the Diffusion Model by Proximal Initialization
>
> [4] Carlini N., et al., 2023, Extracting Training Data from Diffusion Models
>
> [5] Zhang et al., 2025 Position: Membership Inference Attacks Cannot Prove That a Model was Trained on Your Data
>
> [6] Das et al., 2025 Blind Baselines Beat Membership Inference Attacks for Foundation Models

---

> > ### Comment · Reviewer_c42K · 2025-08-05
> >
> > Thanks for the detailed rebuttal and the additional results. These have well-addressed my concerns, and I will raise my rating accordingly.

---

> ### Author Response · Authors · 2025-08-04
>
> Dear reviewer,
>
> As you were concerned about scaling to larger and more diverse settings, we have now performed some additional experiments on the full ImageNet dataset. Together with our CIFAR-100 analysis (included in our rebuttal above), we hope that these address your points regarding scalability.
>
> Scale
> -
>
> In an effort to address the concern regarding large scale experiments, **we have now applied our MIA on an ImageNet diffusion model (over 1m members) using our limited assumptions of 1k samples.** As with the CIFAR-100 experiment we provided above, the external data is chosen from a validation set so we don't expect to see any distribution shifts:
>
> | ImageNet MIA @ 1k budget  | AUC        | TPR @ 1% FPR | ASR     |
> |---------------------------|------------|--------------|---------|
> | Matsumoto et al. [2]      | 50.7       | 0.8          | 50.0    |
> | **Our method**            | **53.0**   | **2.1**     | **51.9** |
>
> Due to time and resource constraints, our method uses only the loss features here. However, the results indicate that even at larger scales trajectory features contain useful information.
>
> **Again, we would like to stress that, to our knowledge, there are no larger scale experiments in the literature in this context. For example, see Table 3 of [1] where their MIAs are only applied models trained on only 8k or 30k samples (and also relies on shadow models). On the other hand, we have thoroughly benchmarked MIAs at 30k (CelebA-HQ) and 50k (CIFAR-10, CIFAR-100) members. While we agree that applying data forensics to large scale systems is of interest, such methods are still far from reliably operating at this scale [3]. We therefore believe that our work demonstrates significant progress in the field that should not be discounted due to scale**.
>
> ---
>
> Once again, we would like to thank you for reviewing our manuscript. We hope that we have resolved your questions and increased your confidence in our work. Please do let us known if there any remaining concerns.
>
> References
> -
>
> [1] Pang Y., et al., 2024, White-box Membership Inference Attacks against Diffusion Models
>
> [2] Matsumoto et al., 2023, Membership Inference Attacks against Diffusion Models
>
> [3] Das et al., 2025 Blind Baselines Beat Membership Inference Attacks for Foundation Models

---

### Official Review · Reviewer_n36r · 2025-07-03

**Clarity:** 4
**Significance:** 3
**Originality:** 3
**Rating:** 5
**Confidence:** 3

**Summary:**

This paper addresses the task for probing data provenance using the whole diffusion trajectories. Specifically, it presents one framework that performs membership inference and model attribution.
Importantly, the paper finds that the “Godlikocks zone” assumption in previous works was not correct. Furthermore, the works explores the effect of distribution shifts on the model performance.
The proposed approach clearly outperforms previous works on various metrics for the considered tasks.

**Questions:**

- Why are foundational features only a problem? It is biased but it might also enable a better performance with a smaller data budget.
- Why is data with high CLIP score a problem?
- Why was the Goldilocks zone adopted? It would be helpful to rephrase the learnings and frame why this might be different to the proposed setting, in what regard the previous learnings were wrong.
- How would the model perform with more / less data budget? Why 3.4%? A sensitivity analysis of this for more experiments would be helpful to better evaluate this.
- What exactly is the model-blind classifier? Please refer or provide more details about it.
- Why only a linear classifier? How would a non-linear model perform?
- Why only such simple projection? Explain why this is reasonable and why more complex projection are not considered!
- What are similarly chosen datasets (l.178)?

**Ethical Concerns:**

["NO or VERY MINOR ethics concerns only"]

**Final Justification:**

I support this work for acceptance since it addresses an important problem in the domain of generative models, it points out incomplete assumptions in previous works and demonstrates its effectiveness in solid experiments.

**Limitations:**

Yes, the limitations are discussed in the main paper.

**Paper Formatting Concerns:**

No major formatting issues noticed.

**Quality:**

3

**Strengths And Weaknesses:**

Strengths

The paper’s motivation is very clear and tackling a very significant topic considering the current developments of diffusion models.
In general, the writing is very clear with the paper having a well-organized structure, where hypotheses are followed by a through experimental analysis. The learnings are subsequently taking into account resulting in clear improvements compared to previous works.
The work does not stick to previous common strategies (Godlikocks zone, no hyperparameter optimization on test set) but instead questions and corrects those.
The work motivates very clean experimental setup removing confounder factors, which increases the confidence of the paper’s learnings.
The model performs well on the considered tasks.

Weaknesses

The access to a few training samples is a strong assumption. The work would benefit from a better sensitivity analysis with respect to the data budget for all performed experiments.
A lower data budget (than 3.4%) would reduce the overall accuracy values, potentially showing clearer gaps for some metrics (e.g. in Tab 5 and 6).
While the work also mentions this as a weakness, it would be more impactful and helpful if the experiments were performed for larger models that are more commonly used.

---

> ### Author Rebuttal · Authors · 2025-07-28
>
> Thank you for the review. We are pleased that you found our setup and presentation clear. We address your questions below:
>
> 1, 2 Why not features from foundation models / CLIP?
> -
>
> (We repeat part of our answer to HK2L, who asked a similar question) **In a security and privacy-critical setting, such as that of data forensics, we care about making predictions based on a minimal set of assumptions**. Foundation models are trained on vast amounts of potentially undisclosed data and there is a risk of data leakage that leads to inflated metrics. In the paper, we give the following example regarding [1], which uses CLIP for black-box Model Attribution (MA) on Stable Diffusion (SD): SD was trained on a filtered version of LAION, where only images with a high CLIP score were included. This means that there are non-trivial implicit dependencies between SD and CLIP. Now, [1] uses CLIP to perform MA on SD, claiming that "only a few images are available", but **CLIP's foundational pre-training being aligned with the diffusion model is an implicit and significant assumption. In contrast, we do not rely on foundation models and make our assumptions explicit in the form of a limited data budget. It is in this sense that we claim that our appraoch is more reliable and our benchmarking more fair.**
>
> 3 Comments on Goldilocks zone hypothesis
> -
>
> **We are happy to further elaborate on this in the final version of the paper.** We include some comments here: Traditionally, in membership inference of discriminative models, a common approach is to infer membership based on thresholding the model's training loss. The Goldilocks zone hypothesis is stated in [2], where the thresholding idea is extended to diffusion models. However, since the true diffusion loss is averaged over all time-steps, for the sake of efficiency, [2] focuses on a single representative time-step. Since then, this approach has, to a large extent, become standard in diffusion model forensics. In contrast, **we show that there is rich information that is encoded in the trajectories that is discarded when performing simple thresholding. Our core argument, which we support with experiments, is that simple thresholding, while fast, is brittle and cannot be trusted in the security and privacy-critical scenarios that we care about in data forensics.**
>
> 4 Effect of the data budget
> -
>
> Please note that **we do have experiments on the effect of the data budget in Figure 5**: we plot performance as a function of the data budget percentage. Note, we consistently outperform the competitors even as the data availability varies. In our experiments, we limit the budget to at most 1000 member samples since we are interested in data forensics under minimal assumptions. For CelebA-HQ this amounts to 3.4%, hence that specific percentage.
>
> 5 Model-blind classifier
> -
>
> As mentioned in the paper, **the model-blind classifier is a neural network that operates directly on raw images, without access to the diffusion models and their features**. In our implementation, **we adopt a standard ResNet18 architecture for this**. The purpose of the model-blind baseline is to quantify the effect of distribution shifts, i.e., the "hardness" of the classification task, and to establish a lower-bound on the performance we would expect from a more sophisticated classifier that also takes into account the model. As you can see from the results **in Table 5, existing MIAs break down in the presence of distribution shifts to the point that this naive baseline outperforms them: this is one of our main arguments against simple thresholding**.
>
> 6, 7 Why a linear model? Why not a more complex network?
> -
>
> We again stress that the aim of data forensics is to make predictions based on a minimal set of assumptions. **In this context, where we assume a limited number of data points for training classifiers, we use a linear model to avoid overfitting**. A linear projection is a simple and objective way to assess the quality of extracted features (see, for example, the vast literature on unsupervised learning). More complex models might require extensive hyperparameter tuning and engineering, which is undesirable. Besides, we note that the simplicity of the linear model allows us to gain some insight on how features are weighted (see Figure 4) thus making the overall pipeline more interpretable.
>
> 8 Similarly chosen datasets
> -
>
> By this we mean that the evaluation set consists of data that has the same distribution and number of samples as the classifier's training set. We have included a detailed explanation of our experimental setup, including the choice of datasets in Appendix A.
>
> A comment on our assumptions and experiments at scale
> -
>
> Here we address comments made in the weakness section of your review. (We repeat part of our answer to 13uc, who had a similar question)
>
> **Assumptions** We claim that **our methodology is more scalable and practical than existing works (e.g. [3] [4]) since we make far milder assumptions in our setup**. For example, as we also mentioned in the paper, **in MIAs it is common to train shadow models to mimic the target DDPM, i.e., assume knowledge of training details of the diffusion models and knowledge of the member distribution to replicate the training**. In contrast, **we avoid this entirely** and make our assumptions explicit in the form of a limited data budget, which is much more realistic and plausible. Of course, if possible, one would rather perform robust MIAs without any assumptions but, to our knowledge, no such method is known (and perhaps this might be theoretically impossible).
>
> **Experiments at larger scale**  To our knowledge, no prior work has rigorously evaluated membership inference at scales beyond what we have tested: results are often reported on subsets of larger datasets. However, in our experiments the diffusion models are trained on the full datasets instead, reflecting a more realistic scenario. For example, Table 3 in [4] shows that their CIFAR-10 DDPM has 8k samples whereas we attack a DDPM trained on the full 50k sample CIFAR-10 dataset.
>
> **As highlighted by recent works, [6], [7], the community is still far from reliably tackling data forensics at the scale of foundation models. Given that even at relatively small scales robust data forensics remains a largely unsolved problem, our work demonstrates significant progress in the field that should not be discounted**.
>
> Still, to further strengthen the paper, we are currently running more experiments. Due to resource and time constraints, it is unlikely that we will be able to complete them during rebuttal. However, we include benchmarking of our method and our strongest competitor, GSA [4], on the full CIFAR-100 dataset below. Specifically, we train the diffusion model on 50k samples of 100 different objects and set the data budget to 2% (1k samples). In this case the external data is chosen from the 10k samples of the validation set of CIFAR-100, so we don't expect to see any distribution shifts. Again, **to our knowledge, our work demonstrates the largest scale MIAs, thoroughly evaluted under a controlled experimental setting with explicit and limited assumptions**:
>
> | CIFAR-100 MIA @ 2% budget | AUC        | TPR @ 1% FPR | ASR     |
> |---------------------------|------------|--------------|---------|
> | Matsumoto et al. [5]      | 58.9       | 1.4          | 56.7    |
> | Pang et al. (GSA) [4]      | 75.7       | 2.7          | 69.8    |
> | **Our method**            | **79.5**   | **10.8**     | **71.6**|
>
>
> We have also benchmarked the performance in the case of object imbalance, simulating a realistic scenario where one may be dealt a member set for training the classifier that is not representative of the member distribution. **In all of these experiments, trajectory features outperform the strongest competitor by a large margin** (please refer to Table 10 and Appendix D for further details and motivation regarding this setting):
>
> | CIFAR-100, TPRs @ 1%FPR | Our method     | GSA        |
> |-------------------------|----------------|------------|
> | Split 1                 | **7.2**        | 3.4        |
> | Split 2                 | **6.1**        | 2.4        |
> | Split 3                 | **7.9**        | 2.5        |
> | Split 4                 | **6.2**        | 3.6        |
> | Split 5                 | **11.4**       | 6.2        |
> | Split 6                 | **5.5**        | 2.1        |
> | Split 7                 | **6.4**        | 3.2        |
> | Split 8                 | **7.1**        | 4.6        |
> | mean ± std              | **7.2 ± 1.85** | 3.5 ± 1.35 |
>
>
> ---
>
> We hope that you found our answers satisfying and we believe that our additional experiments are convicing of the superiority and robustness of our approach compared to existing methods. We remain open to discussion if there are any further clarifications needed.
>
> References
> -
>
> [1] Liu et al., 2025, Which Model Generated This Image? A Model-Agnostic Approach for Origin Attribution
>
> [2] Carlini N., et al., 2023, Extracting Training Data from Diffusion Models
>
> [3] Kong F., et al., 2024, An Efficient Membership Inference Attack for the Diffusion Model by Proximal Initialization
>
> [4] Pang Y., et al., 2024, White-box Membership Inference Attacks against Diffusion Models
>
> [5] Matsumoto et al., 2023, Membership Inference Attacks against Diffusion Models
>
> [6] Zhang et al., 2025 Position: Membership Inference Attacks Cannot Prove That a Model was Trained on Your Data
>
> [7] Das et al., 2025 Blind Baselines Beat Membership Inference Attacks for Foundation Models

---

> > ### Comment · Reviewer_n36r · 2025-08-04
> > **Thank you for the rebuttal !**
> >
> > Thank you for the rebuttal and for clarifying the points that were unclear to me! I appreciate the clear and clean response. It keeps my positive impression of the paper and the thorough work of the authors.
> >
> > Two follow up points:
> >
> > 4) I saw the results in Tab. 5 previously. However, as mentioned in the review, I would encourage to show such evaluations for more experiments. In general, I would also start reporting at 0 ratio and surely also above the selected 3.4%. If the authors are not able to provide this before the discussion ends, I would motivate the authors to report the results in an updated manuscript. I assume, the trend will hold but it will make the claims stronger.
> >
> > Thank you for providing additional results. Since various reviewers pointed out the missing scale, I would encourage the authors to share the updated results on ImageNet in case they still finish during the discussion period?
> > However, I would be supportive of the work even if the findings are not demonstrated at large scale since the trend holds for two datasets.

---

> ### Author Response · Authors · 2025-08-04
> **Thank you for the review!**
>
> Dear reviewer n36r,
>
> We comment on your follow up points below.
>
> Data budget
> -
>
> Focusing on the existing results of Figure 5, we chose to report performance on moderately small ratios for two reasons:
>
> 1. *Too large ratios are not representative of real use cases*. In the context of data forensics we want to make predictions while minimizing our assumptions. Therefore we are mostly interested in the low budget regime.
>
> 2. *The problem is much harder and noisier for extremely small ratios*. Due to the difficulty of the problem, the performance is close to random for all methods. In this regime it is harder to draw conclusions since models are likely to overfit.
>
> **That being said, we are happy to include further experiments in the final version of the paper.** As you suggest, we can explore different ratios and also repeat on different datasets.
>
> Scale
> -
>
> In an effort to address the concern regarding large scale experiments, **we have now applied our MIA on an ImageNet diffusion model (over 1m members) using our limited assumptions of 1k samples.** As with the CIFAR-100 experiment we provided above, the external data is chosen from a validation set so we don't expect to see any distribution shifts:
>
> | ImageNet MIA @ 1k budget  | AUC        | TPR @ 1% FPR | ASR     |
> |---------------------------|------------|--------------|---------|
> | Matsumoto et al. [2]      | 50.7       | 0.8          | 50.0    |
> | **Our method**            | **53.0**   | **2.1**     | **51.9** |
>
> Due to time and resource constraints, our method uses only the loss features here. However, the results indicate that even at larger scales trajectory features contain useful information.
>
> **Again, we would like to stress that, to our knowledge, there are no larger scale experiments in the literature in this context. For example, see Table 3 of [1] where their MIAs are only applied models trained on only 8k or 30k samples (and also relies on shadow models). On the other hand, we have thoroughly benchmarked MIAs at 30k (CelebA-HQ) and 50k (CIFAR-10, CIFAR-100) members. While we agree that applying data forensics to large scale systems is of interest, such methods are still far from reliably operating at this scale [3]. We therefore believe that our work demonstrates significant progress in the field that should not be discounted due to scale**.
>
> ---
>
> Once again, we would like to thank you for reviewing our manuscript. We hope that we have resolved your concerns and increased your confidence in our work.
>
> References
> -
>
> [1] Pang Y., et al., 2024, White-box Membership Inference Attacks against Diffusion Models
>
> [2] Matsumoto et al., 2023, Membership Inference Attacks against Diffusion Models
>
> [3] Das et al., 2025 Blind Baselines Beat Membership Inference Attacks for Foundation Models

---

> > ### Comment · Reviewer_n36r · 2025-08-06
> >
> > Thank you for your efforts!
> > I appreciate your comments. However, I would strongly support to also evaluate in the mentioned regimes - particularly to support these hypotheses. The randomness can, e.g., be evaluated with various seeds. Showing that a method is robustly performing a different method at various scales is very important for this field in my opinion, since the known data ratio is not known.
> >
> > Thanks for the larger dataset evaluation. While they seem not to be common practice, I would argue for not necessarily sticking / constraining oneself to such evaluations but going beyond to increase the impact. Similarly, the Goldilocks zone hypothesis has also proven to be wrong.
> >
> > That being said. I keep my positive rating and increase my confidence.

---

### Official Review · Reviewer_13uc · 2025-07-04

**Clarity:** 3
**Significance:** 3
**Originality:** 3
**Rating:** 4
**Confidence:** 3

**Summary:**

The authors propose a method to determine membership and/or attribution of images to diffusion models. The method extracts features from the entire diffusion trajectory, including loss values, input gradients, and parameter gradients across all time steps, then trains linear classifiers on these compressed representations. This “trajectory feature” departs from the popular “Goldilocks-zone” belief that only mid-range timesteps are useful.

With white-box access and <3.4 % of the training data, the method (i) outperforms prior loss-based and gradient-based MIAs on CIFAR-10 and CelebA-HQ, (ii) stays robust when external data are distribution-shifted, and (iii) can attribute samples from other generators.

The demonstration of the method is limited to small datasets like CIFAR and CelebA.

**Questions:**

Please look at the weaknesses section above.

**Ethical Concerns:**

["NO or VERY MINOR ethics concerns only"]

**Final Justification:**

I was concerned about the scalability of the method, among other concerns. This concern has been alleviated with some of the new experiments added by the authors.

**Limitations:**

yes

**Quality:**

3

**Strengths And Weaknesses:**

## Strengths
1. First of all, this paper does a great job of showcasing that current threshold based MIAs have way too many false positives, and become almost meaningless under distribution shift.
2. The method is able to solve two problems of membership inference and model attribution under the same hood.

## Weaknesses
1.  Experiments are limited to CIFAR-10 and CelebA-HQ-256; no ImageNet-level, text-conditioned, or latent-diffusion models are studied, so real-world generality is unclear.
2. Per-image gradients at hundreds of timesteps are costly. What would the feasibility of this method look like?
3. On CelebA-HQ, attribution accuracy against DDIM is only ~7 % when gradients are used, suggesting poor generalisation to some generators. More analysis of failure cases would be welcome.
4. 3.4% of data is actually quite sizeable as we scale to datasets such as LAION. How would you consider scaling this to larger datasets, and different architectures?
5. I do worry about the generalization of the method to OOD members, or when none of the pool of the çandidate member models is not the one that actually generated the data,. This is another class of false positives,

---

> ### Author Rebuttal · Authors · 2025-07-28
>
> Thank you for the review. We are pleased you appreciated our analysis on pitfalls of existing approaches and that you recognize the value of unifying the various tasks under a single framework. We address your concerns below.
>
> 1 Generality of method and larger scale experiments
> -
>
> **To our knowledge, no prior work has rigorously evaluated membership inference at scales beyond what we have tested**: results are often reported on subsets of datasets. However, in our experiments the diffusion models are trained on the full datasets. For example, Table 3 in [1] (our stongest competitor) shows that their CIFAR-10 DDPM has 8k samples whereas we attack a DDPM trained on the full 50k sample CIFAR-10.
>
> We are running more experiments (DiT, ImageNet). Due to resource and time constraints, it is unlikely that we will be able to complete these during rebuttal. However, we include benchmarking of our method and strongest competitor, GSA [1], on CIFAR-100. Specifically, we train the diffusion model on 50k samples of 100 different objects and set the data budget to 2% (1k). The external data is chosen from the 10k samples of the validation set of CIFAR-100, so we don't expect to see any distribution shifts. Again, **to our knowledge, our work demonstrates the largest scale MIAs, thoroughly evaluted under a controlled experimental setting with explicit and limited assumptions**:
>
> | CIFAR-100 MIA @ 2% budget | AUC        | TPR @ 1% FPR | ASR     |
> |---------------------------|------------|--------------|---------|
> | Matsumoto et al. [2]      | 58.9       | 1.4          | 56.7    |
> | Pang et al. (GSA) [1]      | 75.7       | 2.7          | 69.8    |
> | **Our method**            | **79.5**   | **10.8**     | **71.6**|
>
>
> We have also benchmarked performance in the case of object imbalance, simulating a realistic scenario where one is dealt a member set for training the classifier that is not representative of the member distribution. **In all of these experiments, trajectory features outperform the strongest competitor by a large margin** (see also Table 10 and Appendix D for further details and motivation regarding this setting):
>
> | CIFAR-100, TPRs @ 1%FPR | Our method     | GSA        |
> |-------------------------|----------------|------------|
> | Split 1                 | **7.2**        | 3.4        |
> | Split 2                 | **6.1**        | 2.4        |
> | Split 3                 | **7.9**        | 2.5        |
> | Split 4                 | **6.2**        | 3.6        |
> | Split 5                 | **11.4**       | 6.2        |
> | Split 6                 | **5.5**        | 2.1        |
> | Split 7                 | **6.4**        | 3.2        |
> | Split 8                 | **7.1**        | 4.6        |
> | mean ± std              | **7.2 ± 1.85** | 3.5 ± 1.35 |
>
> Note, even in these experiments, which are relatively small scale compared to something like LAION, it is clear that the data forensics tasks are far from solved and still remain highly non-trivial. **Given that even at these scales robust data forensics remains a largely unsolved problem, our work demonstrates significant progress in the field that should not be discounted**.
>
> **Comments on diffusion variants** As mentioned in the paper, our focus on standard diffusion was intentional: it is the most widely studied and reproducible baseline. Extending to latent/conditional models introduces confounding variables (e.g., VAEs, text encoders), which should be carefully treated to ensure a clean ablation and proper benchmarking. As standard diffusion remains at the core of these more advanced constructions, we believe our analysis and insights remain highly relevant.
>
> 2 Feasibility of method / feature extraction
> -
>
> **We provide discussion on complexity of our method in Appendix C of the paper**. We elaborate in more detail below:
>
> As we acknowledge in our limitations, our trajectory features are more expensive than simple thresholding. Despite this, features can be extracted in a reasonable amount of time, even with a naive sequential implementation that iterates over each sample and time-step one at a time (e.g. ~29s to sequentially extract all features for a single image on an L40S GPU). **To accelerate inference, it is also possible to parallelize feature extraction, as we hint in Algorithm 1 of the paper**. Specifically, **PyTorch supports vectorizing per-sample gradients (e.g. with vmap) and one can parallelize over both time-steps and samples**. This is not something that we have explored in detail because inference time is not a priority. Ultimately, our position is that **real-time application is not critical in the tasks we consider and robustness is a far more important metric: one would rather correctly infer membership in 30s than make a false accusation instantly**. As we show in the paper, existing methods are brittle in practical scenarios and the robustness gains of our method more than make up for the overhead.
>
> As a side note, one can subsample the trajectories (shown in Tables 5,6,7 and Figure 7), to explore a performance-time trade-off.
>
> 3 Model Attribution (MA) generalization
> -
>
> **Conditional accuracy does not reflect a classifier's performance**. It is only meaningful to judge performance over all classes and data. For example, an ImageNet classifier that outputs a constant prediction might score perfectly when conditioned on a specific class but will score very poorly overall. Despite fluctuations in conditional accuracy, observe that **our MA with gradients consistently achieves the best overall accuracy. One can not draw conclusions about generalization from conditional accuracy**.
>
> 4 Assumptions and scaling to larger datasets
> -
>
> **Our methodology is more scalable and practical than existing works (e.g. [1] [2], [3]) since we make far milder assumptions in our setup**. For example, as we mention in the paper, **in MIAs it is common to train shadow models to mimic the target DDPM, i.e., assume knowledge of training details of the diffusion models and knowledge of the member distribution to replicate the training.** In contrast, **we avoid this** and make our assumptions explicit in the form of a limited data budget, which is much more realistic and plausible. Of course, one would rather perform robust MIAs without assumptions but, to our knowledge, no such method is known (and perhaps this might be theoretically impossible).
>
> **Scaling to LAION?** As explained in point 1. of our answers, it is our position that robust data forensics, even in relatively small scale settings, is a largely unsolved problem. Again, **to our knowledge, there is no existing method that has been demonstrated to reliably operate at scales beyond what we have tested. We repeat that a rigorous data forensics analysis is lacking in the literature even in relative small scale settings such as the full CIFAR-10 dataset** (again, it is standard in the literature to consider diffusion models trained on a small subset of the data, see Table 4 of [1]).
>
> **As highlighted by recent works, [3], [4], the community is still far from reliably tackling data forensics at the scale of foundation models.**
>
> **That being said, when dealing massive datasets, a reasonable assumption is that we only care about a small subset at a time**. For example, an artist wishes to perform data forensics on a model trained on vast amounts of data but it is only a small percentage of that data that is of interest to them (e.g. art in their style or related data points). In such cases, one may condition on this subset to perform data forensics.
>
> 5 Some more comments on generalization
> -
>
> We are not sure what you mean by your last question 5 listed under weaknesses. It seems you are asking two separate things.
>
> **OOD members in MIAs** Consider a scenario where the diffusion model is trained on a diverse set of images (e.g. 10 or 100 object classes like CIFAR-10/100) and where the available members for developing our MIAs are limited and may contain class imbalances. For example, focusing on CIFAR, maybe we are dealt a batch of data that is mostly dogs and cats but there are no trucks or airplanes. If we understood your question correctly, you are concerned about such settings where the MIA makes prediction about potentially unseen or under-represented objects. **We have considered such a setting in our CIFAR-10 experiments, shown in Table 10 of Appendix D. We have also included results on CIFAR-100 in our rebuttal point 1. In all these experiments, we outperform the strongest competitor by a large margin.**
>
> **What happens when none of the models is responsible for the data?** We assume you're asking about MA of unseen non-belonging data. Note, in Table 6 our methods have only been trained on a budget of belonging samples (generated by DDPM, always available because white-box access is assumed). Samples from the other models are unseen during training of the classifiers. **Table 6 shows that, on average, our MA classifies generated data from other models (all unseen during training) correctly as non-belonging.**
>
> ---
>
> We hope our responses have addressed your concerns and that our additional experiments are convicing of the superiority and robustness of our approach compared to existing methods. Let us known if further clarifications are needed. Please consider raising your score if you find the above answers sufficient.
>
> References
> -
>
> [1] Pang Y., et al., 2024, White-box Membership Inference Attacks against Diffusion Models
>
> [2] Matsumoto et al., 2023, Membership Inference Attacks against Diffusion Models
>
> [3] Kong F., et al., 2024, An Efficient Membership Inference Attack for the Diffusion Model by Proximal Initialization
>
> [4] Zhang et al., 2025 Position: Membership Inference Attacks Cannot Prove That a Model was Trained on Your Data
>
> [5] Das et al., 2025 Blind Baselines Beat Membership Inference Attacks for Foundation Models

---

> ### Author Response · Authors · 2025-08-04
>
> Dear reviewer,
>
> As you were concerned about scaling to larger and more diverse settings, we have now performed some additional experiments on the full ImageNet dataset. Together with our CIFAR-100 analysis (included in our rebuttal above), we hope that these address your points regarding scalability.
>
> Scale
> -
>
> In an effort to address the concern regarding large scale experiments, **we have now applied our MIA on an ImageNet diffusion model (over 1m members) using our limited assumptions of 1k samples.** As with the CIFAR-100 experiment we provided above, the external data is chosen from a validation set so we don't expect to see any distribution shifts:
>
> | ImageNet MIA @ 1k budget  | AUC        | TPR @ 1% FPR | ASR     |
> |---------------------------|------------|--------------|---------|
> | Matsumoto et al. [2]      | 50.7       | 0.8          | 50.0    |
> | **Our method**            | **53.0**   | **2.1**     | **51.9** |
>
> Due to time and resource constraints, our method uses only the loss features here. However, the results indicate that even at larger scales trajectory features contain useful information.
>
> **Again, we would like to stress that, to our knowledge, there are no larger scale experiments in the literature in this context. For example, see Table 3 of [1] where their MIAs are only applied models trained on only 8k or 30k samples (and also relies on shadow models). On the other hand, we have thoroughly benchmarked MIAs at 30k (CelebA-HQ) and 50k (CIFAR-10, CIFAR-100) members. While we agree that applying data forensics to large scale systems is of interest, such methods are still far from reliably operating at this scale [3]. We therefore believe that our work demonstrates significant progress in the field that should not be discounted due to scale**.
>
> ---
>
> Once again, we would like to thank you for reviewing our manuscript. We hope that we have resolved your questions and increased your confidence in our work. Please do let us known if there any remaining concerns.
>
> References
> -
>
> [1] Pang Y., et al., 2024, White-box Membership Inference Attacks against Diffusion Models
>
> [2] Matsumoto et al., 2023, Membership Inference Attacks against Diffusion Models
>
> [3] Das et al., 2025 Blind Baselines Beat Membership Inference Attacks for Foundation Models

---

### Official Review · Reviewer_HK2L · 2025-07-15

**Clarity:** 2
**Significance:** 2
**Originality:** 3
**Rating:** 4
**Confidence:** 4

**Summary:**

The paper introduced a method for “Origin Attribution”, the goal of determining an image’s origin as one of (a) the training set, (b) generate by the model, (c) other/external. This goal is a generalization of two tasks: Membership Inference Attacks and Model Attribution. The introduced methods consists of exacting features from a given image that consist of a DDPM model gradients for that images over many time steps, and then training a linear probe on top of these features. The authors further study diffusion model dynamics, such as by exploring which time steps lead to features that are most valuable for MIAs.

**Questions:**

Can the authors address the issues mentioned in the "Weaknesses" section? I would be happy to raise the score if these weaknesses were sufficiently addressed.

**Ethical Concerns:**

["NO or VERY MINOR ethics concerns only"]

**Final Justification:**

Thank you to the authors for their response. I raised the score.

**Limitations:**

yes

**Quality:**

3

**Strengths And Weaknesses:**

Strengths
+ The general task of OA makes sense, though it would benefit from further motivation.
+ The proposed method is easy to understand
+ The results for MIA compare favorable to other works.
+ I particularly enjoyed the motivation of the analysis for the “goldilocks zone” hypothesis and the insights drawn from it.

Weaknesses
- The paper has no related works section. While some related works are mentioned throughout (especially in intro and discussion), it would be highly valuable to have a standalone related works section that can provide further context. I’m sure the authors are concerned about page limit, so it should be ok (though probably not ideal) to put related works in the Appendix.
- The authors mention: "We address shortcomings of Wang et al. (2023); Laszkiewicz et al. (2024) which rely on model or layer inversion that are not easily adapted for diffusion. To our knowledge, we are first to demonstrate MA on DDPMs without assistance from pre-trained foundation models.” I have a few issues with this claim. First, both of these method are applying to diffusion models, so it seems their approach is adaptable to diffusion models (even if it not as natural) - for this the authors can compare to these works to show their approaches’ advantage. Second, based on my understanding, Laszkiewicz et al. (2024) does not use a pre-training foundation model as part of the underlying method (but feel free to correct me). Third, even if all other methods to use assistant from pre-training foundation models, it is not clear to me why this is an issue if these method work, since images foundation models will likely always be available.
- The “robustness” evaluations of other MIAs seem somewhat unfair. The point the authors make is that these methods would categorize data generated by a model as part of their training datasets (as opposed to external). However, this is just using these methods for a purpose they are not meant to be used for. Also, in the case that the diffusion models “memorizes” a training datapoint and regurgitates it during inference, it feels closer to being a member than external.

---

> ### Author Rebuttal · Authors · 2025-07-28
>
> Thank you for the review. We are pleased that you found our method easy to understand and that you enjoyed reading our motivation and analysis challenging the Goldilocks zone hypothesis. Please find answers to your questions below:
>
> 1 Related work
> -
>
> **We have discussed related work in the area of data forensics in Section 2.2 (starting at line 87)**.
>
> For Membership Inference Attacks (MIAs) the majority of the literature is on thresholding approaches and we focus on a subset of representative works (as we argue in the paper, thresholding is suboptimal for our purposes).
>
> For Model Attribution (MA) we have also reviewed related work in the same section but, to our knowledge, no existing work has explored the task in diffusion models (please also see our point 2. below).
>
> Origin attribution is a framework that we introduce so again, to our knowledge, there are no existing works.
>
> **We are open to an extended discussion on related works in the Appendix, as you suggest. If there are certain works that you feel we should discuss in detail, please do let us know and we will update the paper with them.**
>
> 2 Existing methods and use of foundation models
> -
>
> We are not claiming that [1], [2] rely on foundation models, these are two separate sentences. To avoid confusion, we will rephrase in the final version of the paper. We clarify below:
>
> **Applicability to diffusion** To our knowledge, there are no works in the literature that have demonstrated or explored white-box MA directly on diffusion models. As we have also stressed in the paper, **existing methods (such as the above-mentioned) only work for latent diffusion models where attribution is only performed on the VAEs and not on the underlying diffusion models**. For example, such approaches completely fail if two latent diffusion models share the VAE (as was done with early versions of Stable Diffusion). Importantly, both of **the above-mentioned works are based on the ideas of model/layer inversion. Such inversion/reconstruction-based methods are fundamentally doomed since zero reconstruction loss is always possible in diffusion [3] (see also interpretation of diffusion as a flow model).**
>
> **Regarding the use of foundation models**: **In a security and privacy-critical setting, such as that of data forensics, we care about making predictions based on a minimal set of assumptions**. Foundation models are trained on vast amounts of potentially undisclosed data and there is a risk of data leakage that leads to inflated metrics. We repeat the example we give in the paper regarding [4], which uses CLIP for black-box MA on Stable Diffusion (SD): SD was trained on a filtered version of LAION, where only images with a high CLIP score were included. This means that there are non-trivial implicit dependencies between SD and CLIP. Now, [4] uses CLIP to perform MA on SD, claiming that "only a few images are available", but CLIP's **foundational pre-training being aligned with the diffusion model is an implicit and significant assumption. In contrast, we do not rely on foundation models and make our assumptions explicit in the form of a limited data budget. It is in this sense that we claim that our appraoch is more reliable and our benchmarking more fair.**
>
> 3 Robustness of MIAs
> -
>
> We respectfully disagree with the comment that MIAs are not meant to distinguish between members and belonging samples. As motivated in the paper, the purpose of data forensics such as MIAs is to empower data owners to hold model developers accountable. For example, an artist wants to make sure that their work was not used to train the model. If an artist's work is used for this purpose, **memorized samples should not be confused with "original" creations by the model. Instead, memorization should be flagged as a data rights violation. Conversely, there is an element of originality in diffusion models and it is therefore not accurate to claim that all generated samples are copies of the training data. In this sense, our experiments on robustness to belonging data reveal a critical flaw of existing MIA frameworks that limits their applicability to the real world (e.g. searching for memorization or auditing belonging data).**
>
> ---
>
> We hope our responses have addressed your concerns and we remain open to further discussion if there are any further clarifications needed. Please consider raising your score if you find the above answers sufficient.
>
>
> References
> -
>
> [1] Wang Z., et al., 2023, Where Did I Come From? Origin Attribution of AI Generated Images
>
> [2] Laszkiewicz et al., 2024, Single-Model Attribution of Generative Models Through Final-Layer Inversion
>
> [3] Huberman-Spiegelglas et al., 2024, An Edit Friendly DDPM Noise Space: Inversion and Manipulations
>
> [4] Liu et al., 2025, Which Model Generated This Image? A Model-Agnostic Approach for Origin Attribution

---

> ### Author Response · Authors · 2025-08-07
> **Follow-up on Rebuttal – Discussion Closing Soon**
>
> Dear Reviewer HK2L,
>
> As the discussion phase is ending soon, we wanted to check whether our rebuttal addressed your concerns. The other reviewers found our clarifications helpful, and we’d greatly value your thoughts as well.

---

### Decision · Program_Chairs · 2025-09-17

**Decision:**

Accept (poster)

**Comment:**

This paper showed that one can use features extracted from the sampling trajectories of a DDPM model to address the task of "origin attribution": i.e. distinguishing whether a given image came from (a) the training set, (b) generate by the model, (c) other/external. The reviewers all felt that the paper was worth of acceptance because (1) the solution is simple & effective, (2) it challenges the prevailing belief that only part of the trajectory is useful for origin attribution (the "goldilocks zone") and (3) the problem is important. While I agreed with the reviewers on these points, I did feel like the paper could have put some more detail into why the features they've selected are the best features for the task. The loss features are motivated as a natural generalization of the prior thresholding methods (which seems reasonable), but the gradient features were less obvious (they work and the curvature story is okay - but we could surely think of other features of the trajectories, and it's not at all obvious that gradients are the right ones to choose). That said, I agree with the reviewers that the paper is interesting and worth appearing at the conference.